# Femtosecond Laser Treatment for Improving the Corrosion Resistance of Selective Laser Melted 17-4PH Stainless Steel

**DOI:** 10.3390/mi13071089

**Published:** 2022-07-09

**Authors:** Lingjian Meng, Jiazhao Long, Huan Yang, Wenjing Shen, Chunbo Li, Can Yang, Meng Wang, Jiaming Li

**Affiliations:** 1Sino-German College of Intelligent Manufacturing, Shenzhen Technology University, Shenzhen 518118, China; 2110412020@stumail.sztu.edu.cn (L.M.); 2110412018@stumail.sztu.edu.cn (J.L.); shenwenjing@sztu.edu.cn (W.S.); lichunbo@sztu.edu.cn (C.L.); yangcan@sztu.edu.cn (C.Y.); wangmeng@sztu.edu.cn (M.W.); 2Guangdong Provincial Key Laboratory of Nanophotonic Functional Materials and Devices, School of Information and Optoelectronic Science and Engineering, South China Normal University, Guangzhou 510631, China; jmli@m.scnu.edu.cn

**Keywords:** additively manufacture, 17-4PH stainless steel, femtosecond laser, corrosion resistance

## Abstract

Currently, laser surface treatment (LST) is considered the most promising method available within the industry. It delivers precise control over surface topography, morphology, wettability, and chemistry, making the technique suitable for regulating the corrosion behavior of alloys. In this paper, femtosecond laser texturing with different parameters and atmosphere environments was adopted to clarify the effect of surface treatment on the corrosion resistance of selective laser melted (SLM-ed) 17-4PH stainless steel (SS) in a NaCl solution. The experimental results show that, after the heat treatment, the corrosion resistance of the laser-treated samples was enhanced. With the further laser treatment in an argon atmosphere, the oxidation of nanostructural surfaces was avoided. The Cr, Cu, and other alloying elements precipitated on the laser-ablated surface were beneficial to the formation of a passivation film, leading to an improved corrosion resistance performance.

## 1. Introduction

Precipitation hardening martensitic 17-4PH stainless steel (SS), due to its excellent mechanical properties and corrosion resistance, is widely used in various fields, such as aerospace, offshore platforms, and nuclear power plants [1]. Metal corrosion can destroy the strength of metal components, leading to the loss of reliability and safety, substantial economic losses, and even catastrophic accidents [2]. To achieve high reliability in various complex working environments, the processed parts are required to have good corrosion resistance [3].

Additive manufacturing (AM) is a layer-by-layer advanced manufacturing process that has emerged as a powerful means for producing metal parts in recent years [4]. This technology can use 3D computer-aided design (CAD) files to fabricate metal parts with complex geometries while saving time and avoiding waste [5,6]. As a typical AM technique, selective laser melting (SLM) can directly prepare high-density metal parts from micro-sized powders without post-processing [7]. AM has been proved to be a very effective and flexible technology for manufacturing high-performance metals, additionally making this process beneficial to corrosion resistance. Adrien Barroux et al. studied the corrosion resistance of 17-4PH SS produced by laser beam melting (LBM) [2] and found that, compared with the forged 17-4PH SS, the LBM-ed specimen had fewer metastable craters, a higher nucleation rate, and a longer service life [8]. Existing research is based on manufacturing processes to improve the corrosion resistance of metals. It is of great significance to develop an effective post-processing process for improving the corrosion resistance of 17-4PH SS.

With characteristics making it highly productive, contactless, and fully automatic, laser polishing is vigorously developed in order to improve surface roughness and to regulate the properties of SLM-ed metals [9]. Lan Chen et al. found that with the combination of surface roughness reduction and grain refinement, the surface properties and electrochemical corrosion behavior of 316L laser-clad SS could be improved effectively by laser polishing [10]. In addition, the laser can also prepare periodic micro-nanostructures on metals to control the surface wettability [11,12]. Superhydrophobic (SH) surfaces with low adhesiveness could be prepared by chemically modifying the laser textured surface [13,14]. Related reports showed that SH surfaces could obtain enhanced passivity, lower anodic dissolution, and corrosion current reduction [15].

Compared with conventional continuous and long-pulse lasers, the ultra-short pulse characteristics of femtosecond lasers can significantly reduce thermal effects during processing and avoid recast layers [16]. In this paper, SLM-ed 17-4PH SS was treated by a femtosecond laser with different parameters in order to regulate the surface structures and corrosion resistance behaviors. The surface characteristics of samples under different processing parameters and their effects on corrosion resistance were studied. The microstructure and phase composition of the laser-treated 17-4PH SS were investigated by optical microscopy, scanning electron microscopy (SEM), energy-dispersive X-ray spectroscopy (EDS), and X-ray diffraction (XRD). Furthermore, the corrosion behaviors were evaluated in 0.5 mol/L NaCl solution at a static temperature of 25 °C, and the passivation film components were assessed using X-ray photoelectron spectroscopy (XPS).

## 2. Materials and Methods

### 2.1. Sample Preparation

The experimental materials—gas-atomized 17-4PH SS powders—were provided by Nantong Jinyuan Intelligence Manufacturing Technology Co. Ltd (Nantong, China). The particle size range of the powders was 15–53 μm, with an average diameter of 32.85 μm, and the standard deviation was 13 μm (Figure 1a). The scanning electron microscopy (SEM) image shows that the powder (Figure 1b) is almost spherical, making it conducive to SLM forming. The chemical composition of the 17-4PH SS powder is shown in Table 1. The SLM-ed cubic 17-4PH steel with dimensions of 15 × 15 × 15 mm^3^ was fabricated in a nitrogen atmosphere. SLM equipment (SLM-100, Han’s Laser Co. Ltd, Shenzhen, China) containing a 200 W fiber laser was used for this experiment. The laser power used for the SLM process was 180 W with a scanning speed of 800 mm/s, and the layer thickness was about 0.03 mm. During the SLM process, the scanning directions for the adjacent forming layers differed by 67°. Part of the SLM-ed samples was selected for a 0.5 h solution treatment at 1040 °C in a muffle furnace and then air-cooled to 550 °C. After that, the samples were aged for 4 h and then air-cooled to room temperature. The oxide layers on the sample surfaces were removed with 80-mesh sandpaper and then cleaned in an ethanol ultrasonic bath for 15 min. Through the use of the Archimedes method, the SLM-ed parts had a density of approximately 99.8%.

### 2.2. Laser Processing

After the heat treatment, the samples were textured by a femtosecond laser. The schematic diagrams for the laser treatment are shown in Figure 2. A 520 nm femtosecond laser (Spectra-Physics Spirit HE 1040-30-SHG, Boston, MA, USA) with a pulse width of 300 fs at a 250 kHz pulse repetition frequency was focused onto the sample surface by an F-theta lens with a focal spot size of 16 µm in diameter. Before laser treatment, the SLM-ed metal surfaces were pretreated with 80-mesh sandpaper. The optimized processing parameters used for laser treatment are shown in Table 2. The laser processing experiments were conducted both in air and argon atmospheres. To obtain SH surfaces, the laser textured samples were placed on a heated plate to be annealed at 110 °C for 2.5 h. Anhydrous ethanol was dripped onto the surface every 30 min during the annealing process.

### 2.3. Surface Characterization and Electrochemical Analysis

The water contact angles (CAs) were measured with 3 µL distilled water using a video optic contact angle instrument (DATAPHYSICS, OCA 25, Stuttgart, Germany). The surface morphology of the laser textured SS was characterized using a 3D measuring laser microscope (Olympus, OSL 4100) and a field-emission scanning electron microscope (FESEM) (Hitachi, Su 8010, Tokyo, Japan). The sample’s surface was etched with a solution consisting of nitric acid (65%), hydrochloric acid (35%), and distilled water in a ratio of 1:10:10 by volume. The phase composition was analyzed by an XRD. The chemical composition of the passivation film for the 17-4PH SS sample obtained after a 72 h immersion in the 0.5 mol/L NaCl solution was investigated by an XPS (ESCA-Lab 250 XI, Thermo-VG Scientific, Waltham, MA, USA) using Al Kα radiation (1486.6 eV).

The action potential polarization and electrochemical impedance spectroscopy (EIS) were measured in 0.5 mol/L NaCl solution using an electrochemical workstation (AUTOLAB, Herisau, Switzerland). The corrosion behaviors of the 17-4PH SS samples were investigated in 0.5 mol/L NaCl solution at a static temperature of 25 °C. In the three-electrode system, the sample’s exposure area was 0.78 cm^2^. In the potential polarization test, the potential scan range was from −1.5 V to +1.5 V (VS EOC), and the scan speed was 0.5 mV/s. The EIS test was carried out to characterize the samples soaked in 0.5 mol/L NaCl solution for 72 h. After the open circuit potential (OCP) was stable for 1800 s, the impedance test was conducted, and the impedance data were collected. The frequency range of the EIS test was 10^−2^~10^5^ Hz, and the ac signal amplitude was 10 mV. Three electrochemical tests were performed on each group of samples to ensure the repeatability of the test results.

## 3. Results and Discussion

### 3.1. Surface Morphology and Wetting Behavior

Figure 3 shows the surface morphologies and wetting behaviors of the SLM-ed SS samples treated with different parameters (Table 2). The polished original sample showed a smooth surface with only a few defects (Figure 3a), and the initial CA was 77°. Figure 3b shows the surface morphologies of the sample ablated by the femtosecond laser in air (FLAR). To describe the different samples concisely, a series of abbreviations were used (Table 3). The rough surface was covered by a large number of nanoparticles with a size of about 100 nm, and the CA was decreased to 9°. When the femtosecond laser processing was conducted in an argon (FLAN) atmosphere (Figure 3c), the surface with a CA of 10° showed an obvious periodic structure. The structure with a period of about 300 nm was perpendicular to the laser polarization direction. This periodic structure may be attributed to the interference between the plasma and the incident laser [17,18]. Moreover, argon can avoid the oxidation of materials during femtosecond laser texturing. After the laser processing in air and argon atmospheres, the surface roughness (Sa) was reduced from 1.709 μm to 1.299 μm and 1.108 μm (Figure 3e).

Preparing a periodic micro-nanostructure is usually necessary to obtain an SH surface. As shown in Figure 3d, with the increased laser fluence and scanning spacing, micro-scale grooves with a width of 10 μm were observed on the laser-ablated surface (Figure 3d). The textured surface manifested superhydrophilicity with a CA of nearly 0°. After the annealing process, the SH surface showed a high CA of 154° and a sliding angle of 3° (Figure 3d). The advancing and receding CAs were 151.03° and 146.18°, respectively, resulting in a CA hysteresis of 4.85°.

### 3.2. Metallographic Analysis

Figure 4 shows the metallographic images and the XRD patterns of the SLM-ed 17-4PH SS samples before and after heat treatment. Before heat treatment, the metal sample was dominated by columnar body-centered cubic (BCC) ferrite (F) grains, composed of elongated subgrains with different growth directions. The sample without heat treatment was labeled F-sample. After heat treatment, the peak value of α′ (110) was significantly increased. This was because the solution treatment refined and homogenized the microstructure. The large-grain ferrite was transformed into a fine acicular martensite (M) lath. However, more intergranular defects could be observed, and the subgrain was not apparent. The heat-treated sample was labeled M-sample. However, the high-heat treatment-induced transformation from ferrite to acicular martensite had little effect on the wetting behaviors of the SLM-ed SS samples treated with different parameters (Figure 3f).

### 3.3. Electrochemical Analysis

#### 3.3.1. Potentiodynamic Polarization Studies

Figure 5 shows the polarization characteristics of each sample with different processes. The corrosion potential (*E*_corr_) and current density (*i*_corr_) were calculated using the Tafel extrapolation method (Table 4). Typically, a high corrosion potential indicates excellent corrosion resistance. The heat treatment caused the charge in the corrosion potential of the original samples to not be apparent. After laser treatment in argon, the F-sample showed a high corrosion potential. The laser-treated M-sample showed the highest corrosion voltage of −0.3820 V. The heat treatment caused an improvement in the corrosion voltage, which can also be observed in the samples treated in air. However, the oxidized nanoparticles on the surface fell away easily, and the exposed matrix accelerated the corrosion. Therefore, the corrosion voltage was relatively low and unstable. The SH M-sample showed the lowest self-corrosion potential. The reason for this is that the polarization current destroyed the low surface energy property caused by the annealing process; therefore, the material’s wetting behavior changed from SH to superhydrophilic. In general, the increased hydrophilicity can lead to an attenuation in the material’s corrosion resistance.

The corrosion current density (*i*_corr_) belongs to the dynamic category. The smaller the *i*_corr_, the slower the corrosion rate. As shown in Figure 5d, the corrosion current densities of the original and FLAN samples were significantly lower than those of the other two groups. The FLAN M-sample showed the lowest corrosion current density of 0.1346 μA·cm^2^. In addition, the refined grain induced by the transformation from ferrite to acicular martensite is beneficial to the corrosion resistance [19,20].

#### 3.3.2. Electrochemical Impedance Spectroscopic (EIS) Studies

Figure 6 shows the measured and simulated impedance characteristics of the eight samples tested in Figure 5. Capacitive arcs appeared on the Nyquist plots for all the samples, revealing that the corrosion reactions occurred at the SS/electrolyte interfaces. Without the heat treatment, the original sample’s arc radius was significantly larger than that of the other three samples. The FLAR sample showed the smallest arc radius. For the heat-treated samples, the laser texturing in argon resulted in the largest capacitive arc radius, and the SH M-sample showed the smallest arc radius. The capacitive arc radius is an essential parameter for evaluating the corrosion resistance of metal materials [21,22]. The larger the capacitive arc radius, the greater the impedance value of the corrosive ions passing through the material surface, and the better the material’s corrosion resistance. This indicates that laser texturing in argon significantly improved the corrosion properties of the M-sample. The high impedances and phase angles indicate that the formed passivation films were more stable for the original and FLAN samples.

To obtain the detailed characteristics of the passivation films, the two equivalent circuit models shown in Figure 7 were chosen to fit the impedance data. *R_s_* represents solution resistance. The equivalent circuit (EC) shown in Figure 7a was named EC-1, and that shown in Figure 7b was named EC-2. The high agreement between the simulated curves and the experiment results fully verifies the circuit’s validity (Figure 6). *R*_f_ and *Q*_f_ represent passivation film resistance and capacitance, respectively. *R*_ct_ and *Q*_dl_ represent charge transfer resistance and double-layer capacitance. The chi-square values (*χ*^2^) were all less than 0.01. The correspondence between the tested samples and ECs, and the fitting results after 72 h are shown in Table 5.

The results indicate that, after 72 h of immersion, the passivation films of the original, FLAN, and SH F-samples remained intact. A similar phenomenon was also observed on the FLAN and SH M-samples. Among them, the FLAN M-sample possessed the largest *R*_f_ of 0.598 MΩ·cm^2^, indicating its high resistance and corrosion-resistant passivation film. However, the high *Q*_f_ reveals that there may be many defects in the FLAN M-sample.

The EIS test showed that, with a small *R*_f_ and a large *Q*_f_, the fabricated SH surfaces did not show a good anti-corrosion performance. In addition, compared with other samples, the micro-sized groove structure enabled the SH surface to process a higher *Q*_f_ value. The results also show that the passivation films of the FLAR F-sample and M-sample were damaged during the test. This is attributed to the oxide particles formed on the laser textured surface. In NaCl solution, these particles tend to fall off, resulting in the destruction of the passivation film.

#### 3.3.3. XPS Characterization

Figure 8 shows the XPS spectra of the passivation films on different samples. The peak spectra, such as Fe2p, Cr2p, Ni2p, Cu2p, Nb3d, and C1s, were fitted to investigate the surface components. The Fe2p spectra of the eight samples show seven peaks around 706.5 eV, 706.9 eV, 708.1 eV, 709.1 eV, 711.5 eV, 718.9 eV, and 723.7 eV, which are related to Fe, FeO/Fe_2_O_3_, and Fe_3_O_4_. The Cr2p spectrum has six peaks around 573.9 eV, 575.4 eV, 576.1 eV, 577.4 eV, 583.5 eV, and 586.4 eV, which are related to Cr, Cr_2_O_3_, and CrCl_3_/Cr(OH)_3_. The Ni2p spectrum contains three peaks around 851.8 eV, 855.0 eV, and 869.9 eV, corresponding to Ni, NiO, and Ni. Two peaks around 933.2 eV and 952.7 eV can be observed in the Cu2p spectra, corresponding to Cu/CuO and Cu_2_O/Cu/CuO. There are three peaks in the Nb2p spectrum, namely, 202.4 eV, 207.2 eV, and 209.5 eV, which are associated with NbO, NbO/Nb_2_O_5_, and Nb_2_O_5_. The C1s spectrum has four peaks around 284.8 eV, 285.6 eV, 286.5 eV, and 588.6 eV, which are related to C.

As we all know, FeO, Fe_2_O_3_, Cr_2_O_3_, and other metal oxides are the main components of SS passivation films [23,24]. It can be seen that the PLAN samples with relatively high polarization and impedance performances showed high peak values of Fe, Cr, Cu, and Nb oxides. This is because the femtosecond laser ablation can induce the precipitation of Cr, Cu, and Nb elements to form an oxide film on the surface, resulting in increased corrosion resistance. This is consistent with the EIS results. Moreover, since the grain is refined after the heat treatment, these metal oxides tend to grow at the grain boundaries with high Gibbs free energy. The increased boundaries in the unit area are conducive to forming a dense and stable passivation film [25]. The SH surface showed the lowest peak values for all the metal oxides, which is consistent with the above impedance results. Meanwhile, this also further confirms the relatively short corrosion reaction time of the SH surface. The low peak values for the C-O bond may have contributed to the lost superhydrophobicity of the SH samples [26].

#### 3.3.4. Electrochemical Corrosion Morphology

Figure 9 shows the surface morphologies after the impedance tests. The FLAN sample showed the fewest defects. The reason may be that the material surface possessed a dense passivation film, so the destruction speed of Cl^−^ to the passivation film was lower than the passivation film’s repair speed. However, the corrosion pits on the surface of the FLAR sample were very obvious. This can be attributed to the severely destroyed passivation film.

Interestingly, the SH surface that had relatively low polarization and impedance performances showed a good corrosion morphology. On the SH M-sample surface, the corrosion traces were very inconspicuous. Some micro-particles were distributed on the laser-fabricated micro-grooves on the SH F-sample. The EDS test showed that carbonaceous particles were derived from the surrounding air. The surface morphologies reveal that the SH surfaces had high corrosion resistance.

#### 3.3.5. Influence Mechanism of Laser Polishing Treatment on Corrosion Resistance

With the little difference in the corrosion morphologies of the original, FLAR, and FLAN samples, the corresponding corrosion mechanism is explained in Figure 10a. For the SH surface with the special structure, the corrosion mechanism is shown in Figure 10b. At room temperature, the chemical reactions of the 17-4PH SS in 0.5 mol/L NaCl solution are as follows:

Firstly, since the oxidation reaction occurs at the cathode, the metallic elements represented by Fe in the SLM-ed SS samples gradually dissolve into the solution after losing electrons at the anode, and the oxidation reaction occurs near the anode:Cr →Cr^3+^ + 3e^−^(1)
Fe → Fe^2+^ + 2e^−^(2)
Fe →Fe^3+^ + 3e^−^(3)
Cu →Cu^+^ + e^−^(4)
Nb →Nb^2+^ + 2e^−^(5)
Nb →Nb^5+^ + 5e^−^(6)

The reduction reaction occurs at the cathode:O_2_ + 2H_2_O + 4e^−^ → 4OH^−^(7)

Around the anode, the above metal ions react with the Cl^−^/OH^−^ in the NaCl solution to form metal chloride or hydroxide, which causes the metal to continue to dissolve in the solution:Cr^3+^ + 3Cl^−^/OH^−^ → CrCl_3_/Cr(OH)_3_(8)
Fe^3+^ + 3Cl^−^/OH^−^ → FeCl_3_/Fe(OH)_3_(9)
Fe^2+^ + 2Cl^−^/OH^−^ → FeCl_2_/Fe(OH)_2_(10)
Cu + Cl^−^ → CuCl(11)
Nb^2+^ + 2Cl^−^/OH^−^ → NbCl_2_/Nb(OH)_2_(12)
Nb^5+^ + 5Cl^−^/OH^−^ → NbCl_5_/Nb(OH)_5_(13)

Next, these metal compounds and residual oxygen in the water together form stable Cr_2_O_3_ and Cu_2_O oxides on the matrix surface, as shown in the XPS results (Figure 9). The formation of these oxides causes the passivation film to be gradually repaired as follows:4Cr(OH)_3_ + 4Cr +3O_2_ → 4Cr_2_O_3_ + 6H_2_O(14)
4Fe(OH)_3_ + 4Fe + 3O_2_ → 4Fe_2_O_3_ + 6H_2_O(15)
Fe(OH)_2_ + 2Fe + O_2_ → 3FeO + H_2_O(16)
4Cu^+^ + 4e^−^ + O_2_ → 2Cu_2_O(17)
Nb(OH)_2_ + 2Nb + O_2_ → 3NbO + H_2_O(18)
2Nb(OH)_5_ + 4Nb + 5O_2_ → 3Nb_2_O_5_ + 5H_2_O(19)

Due to the potential difference between various elements, many micro-cells appear on the material surfaces at the initial stage. These micro-cells can facilitate the migration of Cr^3+^, Fe^2+^, Fe^3+^, Cu^+^, Nb^2+^, Nb^5+^_,_ and Cl^−^ in the solution, decreasing the stability of the passivation films. The dissolved metal cations are combined with the chloride ions and then rapidly oxidize to form metal oxides such as Cr_2_O_3_, FeO, Fe_2_O_3_, Cu_2_O, NbO, and Nb_2_O_5_ (Figure 8). The metal oxides tend to grow at the grain boundaries with a high density, which is conducive to forming passivation films. The passivation film grown along the dense grain boundaries can block the corrosion of metals, resulting in the inhibition of pitting corrosion [25]. However, if there are defects in the passivation, the Cl^−^ can penetrate the metal matrix, resulting in corrosion pits [27].

The Cr, Cu, and other alloying elements precipitated on the FLAN sample surface can lead to an increased thickness of the passivation film. Moreover, the surface nanostructures may hinder the diffusion of oxides, which is beneficial to the stable formation of passivation films. Therefore, the passivation films on the FLAN samples showed relatively high corrosion resistance in the impedance test (Table 5). On the FLAR samples, the loose oxide particles fell off easily during the corrosion process, resulting in substrate exposure. The passivation film was destroyed too fast to be repaired, so the pitting corrosion on the FLAR samples was very obvious (Figure 9f).

The relaviely weak anti-corrosion performance of the SH samples is attributed to the unique surface structure and wetting property. Due to the surface superhydrophobicity, there was an air film between the NaCl solution and the material surface at the initial stage of the test. Therefore, the NaCl solution could not penetrate the micro-nanostructure. However, as the site in contact with the solution was corroded, the liquid gradually penetrated the rough structure (Figure 10b). After 24 h, the superhydrophobicity was wholly lost, and the material surface exhibited superhydrophilicity. As a result, the actual corrosion reaction time of the SH sample was less than that of the other samples, which may be the main reason for the relatively low resistance of the surface passivation film.

## 4. Conclusions

In this study, the corrosion behaviors of SLM-ed 17-4PH SS treated with different femtosecond laser parameters were investigated.

(1)The F-sample was dominated by columnar BCC ferrite grains composed of elongated subgrains with different growth directions. After the heat treatment, the large-grain ferrite was transformed into a fine acicular martensite lath, resulting in improved corrosion resistance.(2)Femtosecond LST was used to induce periodic nanostructures on the material surfaces. The argon atmosphere effectively prevented the oxidation of the laser-ablated surface. Moreover, the Cr, Cu, and other alloying elements precipitated on the FLAN sample surface were beneficial to the formation of the passivation film, leading to excellent corrosion resistance performance.(3)Since the wetting behavior was transformed from SH to superhydrophilic, the fabricated SH surfaces did not show a good anti-corrosion performance. However, the air film between the solution and the material surface delayed the surface corrosion, resulting in inconspicuous corrosion pits.

## Figures and Tables

**Figure 1 micromachines-13-01089-f001:**
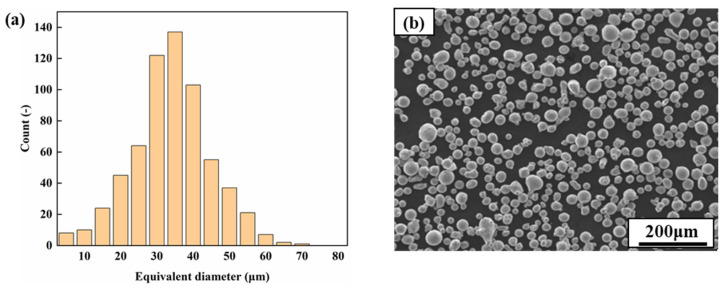
(**a**) Histogram of the powder’s equivalent diameter and (**b**) representative powder image.

**Figure 2 micromachines-13-01089-f002:**
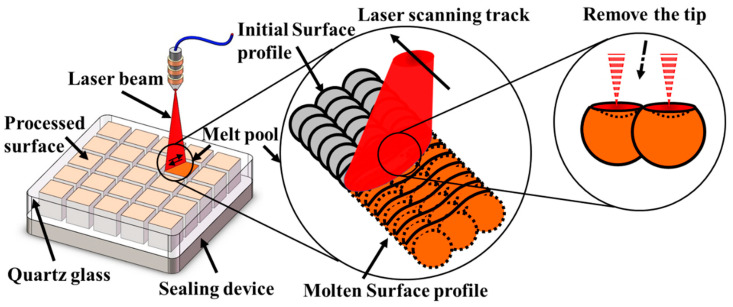
Schematic illustrating the laser surface treatment of the 17-4PH SS.

**Figure 3 micromachines-13-01089-f003:**
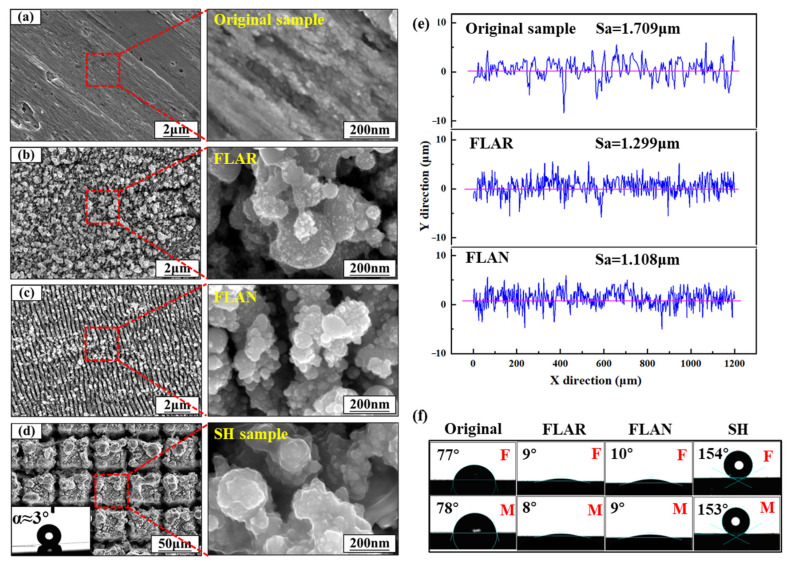
Surface structures of (**a**) original, (**b**) FLAR, (**c**) FLAN, and (**d**) SH samples. (**e**) Cross-section dimensions of original, FLAR, and FLAN samples. (**f**) Wetting behaviors of original, FLAR, FLAN, and SH samples.

**Figure 4 micromachines-13-01089-f004:**
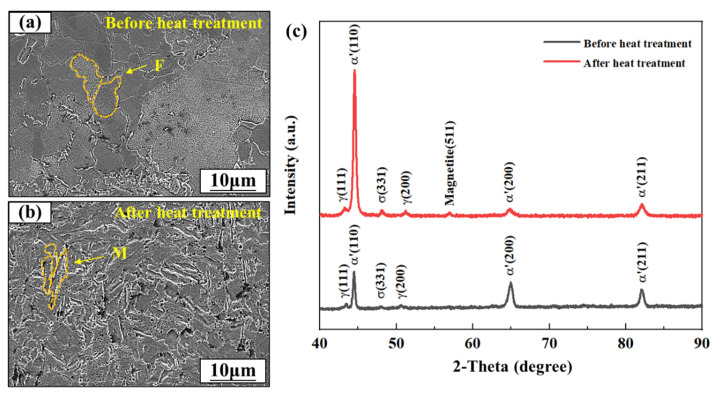
Microstructures of the SLM-ed 17-4PH SS (**a**) before and (**b**) after heat treatment. (**c**) XRD patterns of the SLM-ed 17-4PH SS before and after heat treatment.

**Figure 5 micromachines-13-01089-f005:**
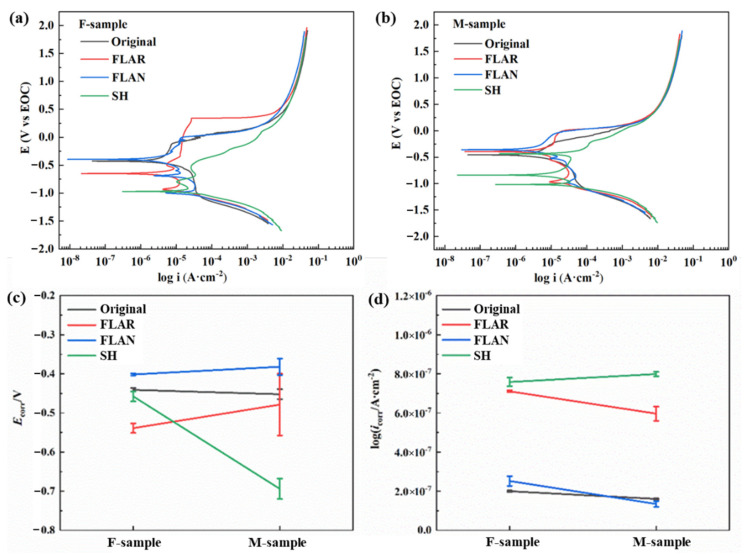
Potentiodynamic polarization curves of the (**a**) F-samples and (**b**) M-samples. (**c**) Corrosion voltages and (**d**) corrosion currents of SLM-ed samples with different post-treatment processes.

**Figure 6 micromachines-13-01089-f006:**
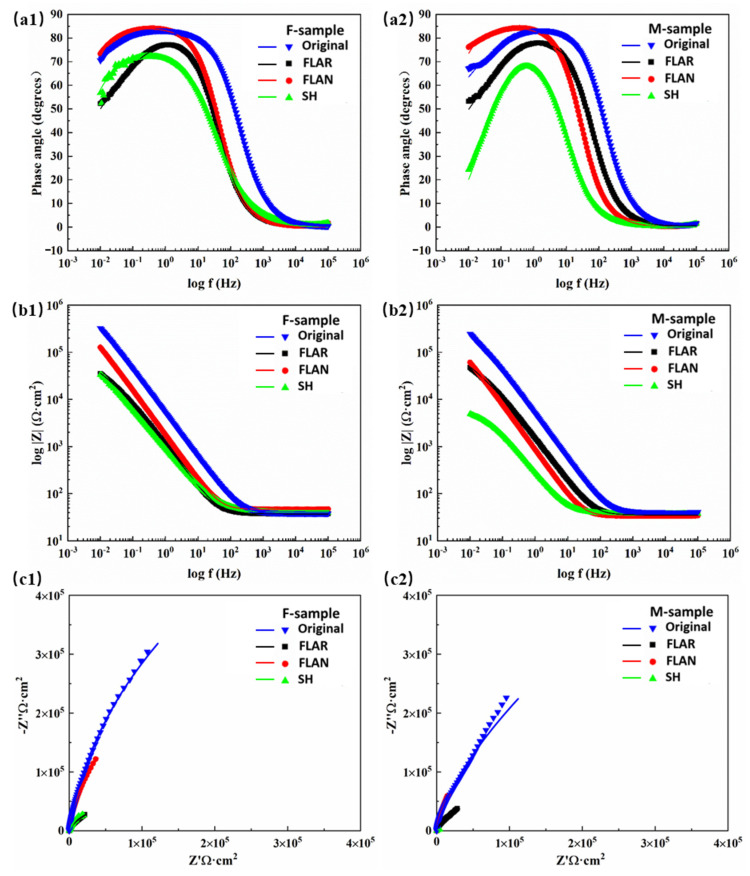
Measured and simulated (**a**) Bode phase angle, (**b**) Bode impedance, and (**c**) Nyquist curves of SLM-ed samples with different post-treatment processes.

**Figure 7 micromachines-13-01089-f007:**
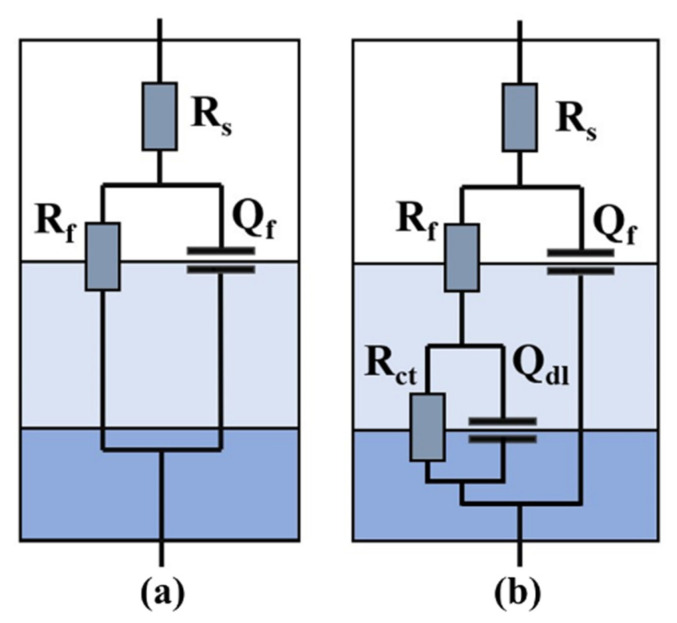
Electrochemical equivalent circuits for fitting the measured impedance data of (**a**) the FLAR samples and (**b**) other samples.

**Figure 8 micromachines-13-01089-f008:**
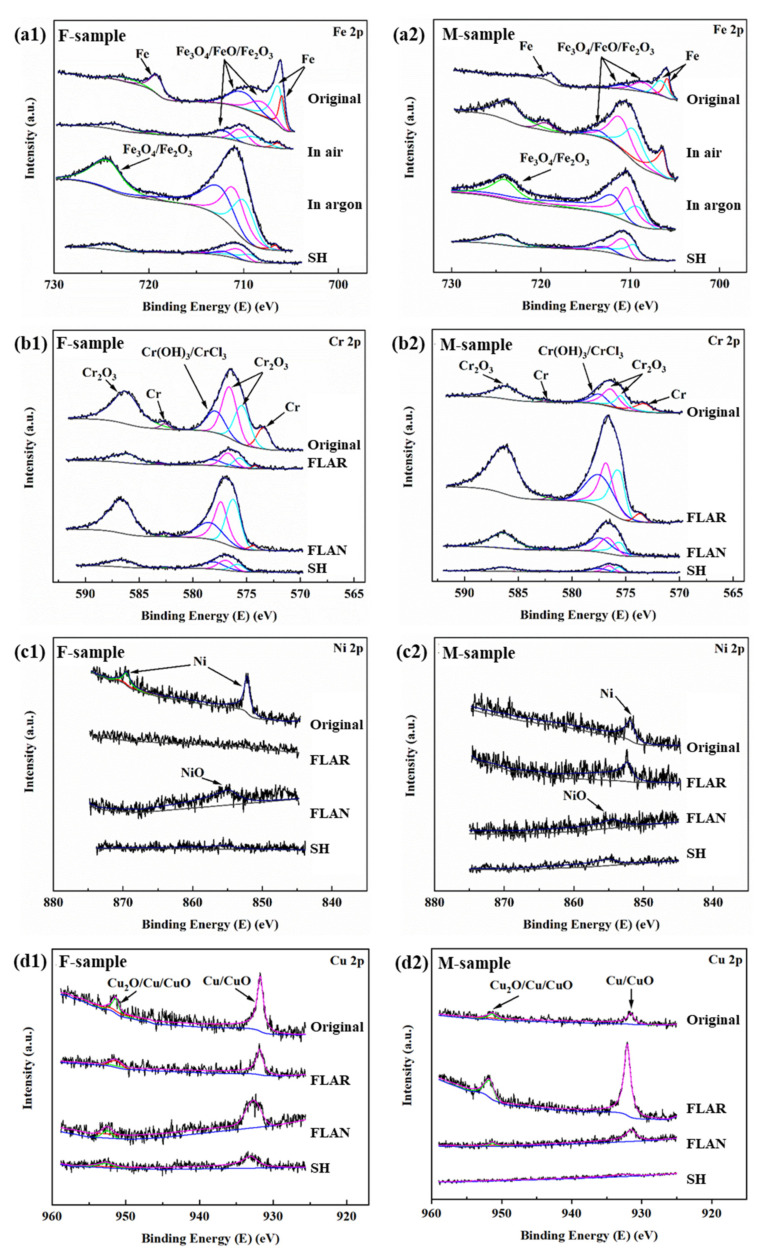
High-resolution XPS spectra of (**a**) Fe 2p, (**b**) Cr 2p, (**c**) Ni 2p, (**d**) Cu 2p, (**e**) Nb 3d, and (**f**) C 1s for SLM-ed samples after being immersed in 0.5 mol/L NaCl solution for 72 h.

**Figure 9 micromachines-13-01089-f009:**
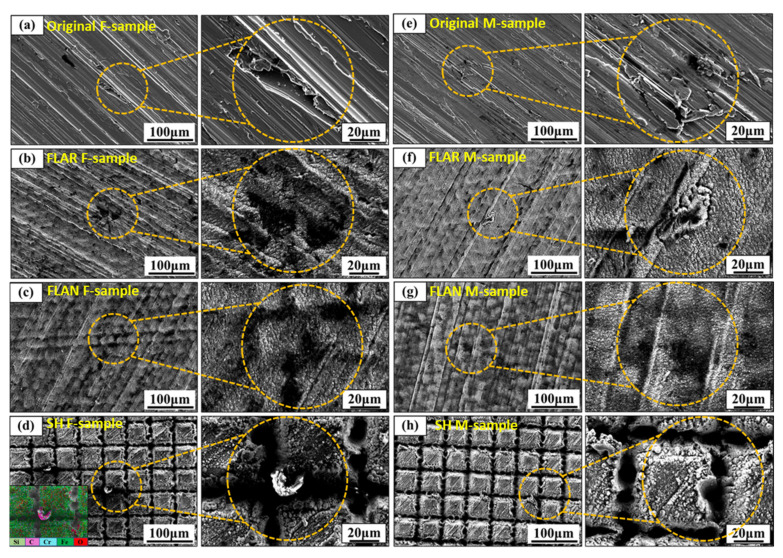
Surface morphologies of (**a**) original F-samples, (**b**) FLAR F-samples, (**c**) FLAN F-samples, (**d**) SH F-samples, (**e**) original M-samples, (**f**) FLAR M-samples, (**g**) FLAN M-samples, and (**h**) SH M-samples after being immersed in 0.5 mol/L NaCl solution for 72 h.

**Figure 10 micromachines-13-01089-f010:**
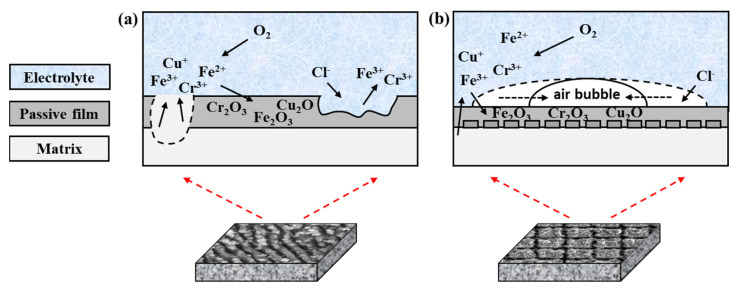
Schematic illustrating the corrosion mechanism of (**a**) hydrophilic and (**b**) SH 17-4PH SS.

**Table 1 micromachines-13-01089-t001:** Chemical composition of 17-4PH SS powder (wt %).

Fe	Cr	Ni	Cu	Mn	Nb	Si	O	C	P	S
balance	17.01	4.69	4.03	0.59	0.34	0.26	0.05	0.04	0.012	0.007

**Table 2 micromachines-13-01089-t002:** Laser processing parameters used in this study.

	Air	Argon	SH
Scanning speed (mm/s)	200	200	100
Processing times (-)	1	1	5
Fluence (J/cm^2^)	1.27	1.27	6.37
Scanning spacing (μm)	2	2	50

**Table 3 micromachines-13-01089-t003:** Sample labels.

Full Name	Femtosecond Laser Treatment in Air	Femtosecond Laser Treatment in Argon	Superhydrophobic	Ferrite Sample	Martensite Sample
Abbreviation	FLAR	FLAN	SH	F-sample	M-sample

**Table 4 micromachines-13-01089-t004:** Quantitative information about the potentiodynamic polarization curves of 8 samples in 0.5 mol/L NaCl solution.

Sample	Average Log *i*_corr_ (μA·cm^−2^)	Average *E*_corr_ (V)
Original F-samples	0.1999 ± 2.0%	−0.4408 ± 1.0%
Original M-samples	0.1606 ± 1.6%	−0.4521 ± 2.7%
FLAR F-samples	0.7107 ± 0.6%	−0.5389 ± 2.2%
FLAR M-samples	0.5963 ± 6.1%	−0.4786 ± 16.5%
FLAN F-samples	0.2518 ± 9.7%	−0.4017 ± 0.6%
FLAN M-samples	0.1346 ± 10.7%	−0.3820 ± 5.5%
SH F-samples	0.7583 ± 2.9%	−0.4575 ± 2.8%
SH M-samples	0.7988 ± 2.0%	−0.6940 ± 3.8%

**Table 5 micromachines-13-01089-t005:** Fitting results of EIS for 8 samples in 0.5 mol/L NaCl for 72 h.

Sample	*R*_f_ (MΩ·cm^2^)	*Q*_f_ (μF·cm^−^^2^)	*R*_ct_ (MΩ·cm^2^)	*Q*_dl_ (μF·cm^−2^)
Original F-samples (EC-1)	0.142 ± 16.9%	35.269 ± 5.3%	0	0
Original M-samples (EC-1)	0.173 ± 23.7%	59.290 ± 10.9%	0	0
FLAR F-samples (EC-2)	0.0000156 ± 26.3%	45.972 ± 14.9%	3.228 ± 6.7%	155.931 ± 12.4%
FLAR M-samples (EC-2)	0.0000166 ± 16.9%	33.936 ± 19.9%	8.049 ± 4.2%	115.330 ± 10.6%
FLAN F-samples (EC-1)	0.313 ± 9.3%	99.970 ± 21.8%	0	0
FLAN M-samples (EC-1)	0.598 ± 5.2%	206.392 ± 11.8%	0	0
SH F-samples (EC-1)	0.126 ± 28.6%	243.610 ± 5.1%	0	0
SH M-samples (EC-1)	0.00562 ± 6.6%	780.84 ± 5.5%	0	0

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
