# Peer review of "Femtosecond Laser Treatment for Improving the Corrosion Resistance of Selective Laser Melted 17-4PH Stainless Steel"

_micromachines, 2022, doi:10.3390/mi13071089_

Round 1
Reviewer 1 Report
This manuscript reports the enhancement of corrosion resistance of 17-4PH stainless steel after femtosecond laser treatment. The authors conducted a very comprehensive characterization of the laser treated metal followed by a qualitative discussion of the mechanism. Overall, the manuscript is well-written, and the finding is very informative. The reviewer only has a few minor comments.
1. What is the rationale behind choosing the current laser processing parameters listed in Table 2?
2. It is better to add the value of surface roughness in Fig. 3e to be more informative.
3. It is better to add a table to summarize the sample labels used in the manuscript. (i.e. FLAR, FLAN, SH) so that the reader can easily find the definition if they get lost.
4. Fig. 5a is hard to read due to too much overlapping lines. The reviewer would suggest separating the F-sample and M-sample into two figures with the same scales.
Reviewer 2 Report
In this work, the authors have performed femtosecond laser texturing on selective laser melted (SLM) 17-4PH stainless steel (SS) parts to study the corrosion resistance. The work contains some interesting results. They found that laser texturing in an argon atmosphere avoided surface oxidation and benefited the formation of passivation film to improve corrosion resistance. The manuscript is well-written and can be published with minor revisions. The reviewer has the following comments.
1. Previous research has shown that laser-induced periodic surface structures (LIPSS) (similar structure like FLAN) can also be superhydrophobic. The reviewer suggests including the contact angle of all 8 conditions. This will provide complete information on how heat treatment influenced wettability.
https://doi.org/10.1021/la2011088
https://doi.org/10.1016/j.apsusc.2009.07.061
2. A recent review article on the role of surface chemistry modification due to heat treatment after laser texturing (https://doi.org/10.1016/j.matdes.2020.108744) on corrosion resistance. Wettability transition from superhydrophilicity (immediately after laser texturing) to superhydrophobicity upon heat treatment is due to accelerated adsorption of hydrocarbon compounds from the environment. The authors have shown XPS results of Fe 2p, Cr 2p, Ni 2p, and Cu 2p. It might be beneficial to include C 1s to show the readers whether the adsorbed carbon content also plays a role in the improved corrosion resistance in the FLAN-M case.
3. Superhydrophobicity cannot be claimed with only static contact angle. Therefore, additional parameters, including roll-off angle, contact angle hysteresis, advancing and receding contact angle, need to be included in the manuscript.
4. The introduction section of the manuscript is weak for a review related to recent works on the effect of laser texturing on wettability and corrosion resistance. It should be improved to establish the focus of this work.
https://doi.org/10.1016/j.corsci.2017.04.005
https://doi.org/10.1016/j.apsusc.2018.06.033
https://doi.org/10.1021/acsami.5b06217
https://doi.org/10.3390/ma11091577
https://doi.org/10.1016/j.apsusc.2019.145136
5. The current manuscript only shows measured EIS results. Please provide simulated EIS curves based on electrochemical equivalent circuits (Figure 7) to verify the circuit's validity.
